# Self-Healing Properties of Asphalt Concrete with Calcium Alginate Capsules Containing Different Healing Agents

**DOI:** 10.3390/ma15165555

**Published:** 2022-08-12

**Authors:** Huoming Wang, Miao Yuan, Jie Wu, Pei Wan, Quantao Liu

**Affiliations:** 1China Merchants Chongqing Communications Technology Research and Design Institute Co., Ltd., Chongqing 400067, China; 2Foshan Traffic Technology Co., Ltd., Foshan 528041, China; 3State Key Laboratory of Silicate Materials for Architectures, Wuhan University of Technology, Wuhan 430070, China

**Keywords:** asphalt mixture, calcium alginate capsules, different healing agents, self-healing

## Abstract

Calcium alginate capsules encapsulating rejuvenator are a promising self-healing technology for asphalt pavement, but the effects of different healing agents on the self-healing performance of asphalt concrete has not been considered. In view of this, this paper aimed at exploring the effects of calcium alginate capsules containing different healing agents on the self-healing properties of asphalt concrete. Three types of capsules with sunflower oil, waste cooking oil and commercial rejuvenator were fabricated via the orifice-coagulation bath method and the interior structure, mechanical strength, thermal stability and oil content of the prepared capsules were characterized. The healing levels of asphalt mixtures with different capsules under different loading cycles and stress levels were evaluated. Furthermore, the saturates, aromatics, resins and asphaltenes (SARA) fractions and rheological property of extracted asphalt binder within test beams with different capsules after different loading conditions were assessed. The results indicated that all the three types of capsules meet the mechanical and thermal requirement of mixing and compaction of asphalt mixtures. The healing levels of test beams containing vegetable oil capsules were higher than that of waste cooking oil capsules and industrial rejuvenator capsules. The strength recovery ratio and fracture energy recovery ratio of test beams with vegetable oil capsules reached 82.8% and 96.6%, respectively, after 20,000 cycles of compressive loading at 1.4 MPa. The fracture energy recovery ratio of the waste cooking oil capsules also reached as high as 90%, indicating that waste cooking oil can be used as the healing agent of calcium alginate capsules to improve the self-healing property of asphalt mixture. This work provides a significant guide for the selection of healing agent for self-healing capsules in the future.

## 1. Introduction

Asphalt concrete has become the dominant surface paving material on high-grade roads due to its superior pavement performance. Nevertheless, ageing and cracking will inevitably occur on asphalt pavement due to low temperature [1], UV exposure [2], moisture weather [3] and traffic loading [4] after serving a period of time. Micro cracks will fatally deteriorate into macro cracks without proper maintenance applied on asphalt pavement, thus not only shortening the service period of pavement, but also affecting the integrity of pavement structure and the vehicle safety. Hence, to maintain asphalt pavements in a good condition during their lifetime, external maintenance is usually applied on pavements by the road maintenance department. The current maintenance activities aiming at repairing the cracks are all passive maintenance measures after macroscopic damage occurs in the pavement, which not only consume masses of natural resources, but also cause severe environmental pollution, such as secondary organic aerosol precursors [5], volatile organic compounds [6,7] and greenhouse gas [8]. Therefore, efficient and clean maintenance technologies are urgently needed for asphalt pavement in the context of low carbon maintenance [9,10].

Asphalt is a kind of self-healing material, and can spontaneously heal the inner micro cracks during rest periods or at high temperatures [11,12]. However, the self-healing efficiency of asphalt pavement is low resulting from low temperature [13] and asphalt ageing [14] during the service condition. In recent years, some researchers have made achievements in improving the self-healing performance of asphalt mixtures [15]. It is proven that asphalt rejuvenator encapsulation method can not only effectively enhance the micro crack-repair efficiency of asphalt mixtures [16,17], but also locally rejuvenate aged asphalt binder due to the timely supply of lost light components [18], thus being a promising pre-maintenance technology for asphalt pavement in the future

The capsules containing asphalt rejuvenator are the main encapsulation form for improving the self-healing levels of asphalt materials (asphalt binder and mortar and mixtures). The self-healing capsule consists of core-shell microcapsules and multi-cavity capsules. The microcapsules with core-shell structure (5~500 μm) are principally synthesized by in-situ polymerization method. Usually, melamine–formaldehyde (MF) [19,20], methanol–melamine–formaldehyde (MMF) [21,22], urea–formaldehyde (UF) [23,24] and melamine–urea–formaldehyde (MUF) [25,26] are selected as shell materials to load asphalt rejuvenator. It has been confirmed that asphalt binder or mortar containing microcapsules have good self-healing performance in the fracture recovery test [27,28]. Nevertheless, asphalt mixtures containing microcapsules failed to effectively improve the healing level of fatigue crack due to limited healing agent and one-time rupture release mode. The multi-cavity capsules (1~10 mm) are fabricated by the orifice-coagulation bath method [29,30,31,32]. Based on reaction principle of ion exchange, calcium alginate is usually used as shell material to coat vegetable oil with masses of light components. It has been proved that asphalt concrete containing calcium alginate capsules showed better performance in multiple crack healing and sustainable healing resulting from sufficient rejuvenator stored in compartmented cavities and gradual release of the healing agent [33,34,35].

It is noting that most of current literatures focus on the fabrication of capsules with sunflower oil as the healing agent and ignore the feasibility of other rejuvenator embedded in calcium alginate capsules. The effects of different healing agents on the self-healing performance of asphalt concrete has not been considered. In view of this, the paper aimed at exploring the self-healing properties of asphalt mixtures with calcium alginate capsules containing different healing agents. First, calcium alginate capsules containing vegetable oil, waste cooking oil and commercial rejuvenator were fabricated, respectively. Secondly, a series of test were conducted to characterize the interior structure, mechanical strength, thermal resistance and relative rejuvenator content of the capsules. Thirdly, the healing levels of asphalt mixtures with the three types of capsules under various cycles of compressive loading and different loading pressure were measured. Moreover, the rheological properties and the SRAR fractions of the extracted asphalt binders within the test beams after compressive loading were explored by Dynamic Shear rheometer (DSR) and thin-layer chromatography-hydrogen flame ion detector respectively.

## 2. Materials and Methods

### 2.1. Materials

The raw materials used to prepare calcium alginate capsules include sodium alginate, surfactant, healing agents, calcium chloride and tap water. Sodium alginate, surfactant Tween 80 and calcium chloride were purchased from Sinopharm Chemical Reagent (Beijing, China). Three kinds of healing agents (vegetable oil, waste cooking oil and industrial rejuvenator) were used as core materials to prepare the capsules. The vegetable oil (rapeseed oil) was purchased from a local supermarket, the waste cooking oil was obtained from a local gutter oil collection canter, the industrial rejuvenator was purchased from Haina Industry and Trade Co., Ltd. (Wuxi, China). The properties of the three healing agents were shown in Table 1.

### 2.2. Preparation of Calcium Alginate Capsules

Calcium alginate capsules containing different healing agents were fabricated by orifice-bath method. Based on previous research [33,35,36], the encapsulation procedure of calcium alginate capsules is shown in Figure 1: (1) 8 g sodium alginate was added into 392 g deionized water and the mixture was stirred until the sodium alginate was completely dissolved. (2) An amount of 40 g healing agent and 2 mL Tween 80 were added to the sodium alginate solution and then sheared for 15 min at 5000 rpm to form an oil-in-water emulsion. (3) The oil-in-water emulsion was dropped into a CaCl_2_ solution with a mass fraction of 3.0% at a speed of 65–70 drops/min and a temperature of 50 °C, forming Ca-alginate cross-linking encapsulating the healing agent inside. The capsules were kept in the CaCl_2_ solution for 4 h to ensure that the Na^+^ in sodium alginate completely reacted with Ca^2+^. (4) The prepared capsules were washed and dried at room temperature to obtain the dried capsules.

### 2.3. Characterization of the Capsules

In this study, the internal structure of the capsules was observed by scanning electron microscopy (SEM) (JSM-7500F, Tokyo, Japan). Firstly, the capsule was cut into semicircles with a blade, then the surface of the capsule was sprayed with gold, and finally, the cross-section of the capsule was scanned with an electron microscope at a voltage of 5 kV. The mechanical strength of the capsules was measured by uniaxial compression test at room temperature using the multi-functional mechanical testing machine (ZQ-990, Wuhan, China) with a loading rate of 0.5 mm/min. As the force increases during loading, a yield point was formed on the stress-displacement curve when the capsule was destroyed, and the force corresponding to the yield point was recorded as the mechanical strength of the capsule. The thermal stability of the capsules was determined by thermogravimetric analysis (TGA) using the synchronous thermal analyzer (STA 4494c/3/G, Selbu, Germany). Moreover, the oil content in the capsules was quantified according to the thermogravimetric curves. The heating range was from 40 °C to 1000 °C and the heating rate was 10 °C/min. Nitrogen was injected as a protective gas throughout the heating process with an inlet rate of 20 mL/min.

### 2.4. Asphalt Specimen Preparation

Unmodified 70# bitumen with a softening point of 48.1 °C, a ductility of 15.5 mm (5 cm/min, 15 °C) and a penetration of 65.3 (0.1 mm) at 25 °C was purchased from Inner Mongolia New Road Asphalt Fuel Distribution Co., Ltd. (Hohhot, China). The aggregates used to prepare asphalt mixtures were basalt aggregates.

A dense graded asphalt mixture AC-13 was used in this study. The aggregate gradation was presented in Figure 2. The optimum asphalt content determined by Marshall design methods was 4.5%. The asphalt mixture designed had an air void (VV) of 4.0%, a bulk density of 2.489 g/cm^3^, void in mineral aggregate (VMA) of 15.6% and void filled with asphalt (VFA) of 74.6%, meeting the requirements of “Technical Specification for Construction of Highway Asphalt Pavement (JTG-F40-2004)” [37].

It is reported that the content of calcium alginate capsules shouldn’t be higher than 0.5% to avoid the adverse effect on the water sensitivity and fatigue resistance of asphalt mixture [36,38]. In this research, calcium alginate capsules with a content of 0.4% were added to asphalt mixture without changing the gradation to study their effects on the self-healing properties of the mixture. Standard rutting plate specimens were prepared using a rutting plate molding roller. Asphalt mixture beams with size 95 mm × 50 mm × 45 mm were obtained from rutting plates and a 10 mm × 4 mm notch was created in the middle of each beam to make sure crack appears at the same position in the following fracture-healing test.

### 2.5. Self-Healing Property Evaluation

The effects of calcium alginate capsules containing different types of healing agents on the self-healing properties of the asphalt specimens were evaluated by fracture-cyclic compression loading–healing test, shown in Figure 3.

The fracture-cyclic compression loading–healing test procedure included four steps: (1) three-point bending test was performed on asphalt specimens at −10 °C with a loading speed of 0.5 mm/min to test their initial flexural strength and generate cracks to heal. (2) The fractured samples were placed into the steel mold and different cycles of cyclic compression loading (0, 5000, 10,000, 15,000 and 20,000) of 2 Hz were applied on the fractured beams to simulate the vehicle tire loading on cracked beams and to force the capsules release the oil under cyclic compression pressure. The loading stress was 0.6 MPa, 1.0 MPa and 1.4 MPa, respectively. (3) After cyclic compression loading, specimens were kept in the steel mold at 20 °C for 3 d to heal. (4) Three-point bending test was repeated on the healed specimens to test their flexural strength after healing as described in step 1.

Two healing indices were used to evaluate the self-healing levels of the specimens. Strength recovery ratio was defined as the yield load of the healed beam (*F*_2_) divided by the initial yield load of the beam (*F*_1_). Fracture energy recovery ratio was defined as the fracture work of the healed specimen (*W*_2_) divided by its initial fracture work (*W*_1_). The fracture work (*W_f_*) can be calculated as the area under force-displacement curve until the yield load according to Equation (1).
(1)$$W_{f}=\int_{0}^{b}Fdu$$
where b is the displacement when the yield force was reached, and u and F are the load line displacement and applied compression loading.

### 2.6. SARA Fractions Analysis

Asphalt binders were extracted from the specimens before and after healing. An Iatroscan MK-6 rod-shaped thin-layer chromatography-hydrogen flame ion detector was used to measure SARA fractions (i.e., Saturates, Aromatics, Resins and Asphaltenes) of asphalt binders before and after healing to analyze the effect of different healing agents released from the capsules on asphalt binder. An amount of 80 mg of the extracted asphalt binder was dissolved in 4 mL of tetrachloromethane solution and 1 μL of the solution were dropped on the chromatographic bar. Then, it was placed in three configured solutions (n-heptane, a mixture of 20% n-heptane and 80% toluene, a mixture of 55% toluene and 45% ethanol) for 30 min, 8 min and 2 min, respectively, to separate the four components of asphalt. Finally, the organic matter detected by the hydrogen flame ion system was quantitatively analyzed.

### 2.7. Rheological Characterization of the Extracted Asphalts

The rheological properties of 70# asphalt and different types of extracted asphalt binders were tested by temperature sweep test. The test was conducted using Dynamic Shear Rheometer (DSR) (Anton Paar, Smartpave 102, Ostfildern-Scharnhausen, Germany) according to AASHTO T315 to measure the complex modulus (G*) and phase angle (δ) of the binders at different temperatures. The temperature sweep test was performed from 40 °C to 80 °C with a temperature increment of 2 °C/min. The strain amplitude was 0.05%. The diameter of the rotor was 25 mm and the gap between the rotor and the plate was 1 mm.

## 3. Results and Discussion

### 3.1. Interior Structure of the Capsules

The rejuvenator storage is related with the interior structure of capsules. Figure 4 presented the typical structure of three types of calcium alginate capsules containing different healing agents. It can be seen that all the capsules owned a muti-cavity structure. The cavities in different capsules showed differences in size and shape, which may be attributed to the viscosity otherness of the emulsions informed by sodium alginate solution and healing agents. Industrial rejuvenator with higher viscosity can form larger oil droplets during the shearing process which results in bigger cavities within the capsule after cross-linking encapsulation. It was noting that the healing agent was stored in different cavities in the capsules rather than embedded wholly in the core-shell structure microcapsules. The unique rejuvenator storage mode can provide sustained rejuvenator release potential of the capsules [31,32,33,34,35,36,38].

### 3.2. Mechanical Strength of the Capsules

In this work, the capsules were added into asphalt mixtures as partial fine aggregates. Hence, the capsules must resist the compression during the mixing and compaction of asphalt mixtures. It has been proven that the strength of capsules added as aggregates in asphalt mixtures should be higher than 10 N to resist the stress generated during the production of asphalt concrete. Figure 5 showed the mechanical resistance of calcium alginate capsules containing different healing agents. The mechanical resistances of rapeseed oil capsule, waste cooking oil capsule and industrial rejuvenator capsule were 13.2 N, 12.7 N and 13.5 N, respectively. It is believed that the slight difference in the mechanical resistance of the three types of capsule was caused by the difference in the cavity size of the capsules. The mechanical resistances of three types of capsules were all higher than the threshold strength, which indicated that the capsules can withstand the stress action during the mixing and compaction of asphalt mixtures.

### 3.3. Thermal Stability and Rejuvenator Content of the Capsules

The capsules were subjected to mixing and compaction at high temperatures. Hence, the decomposition temperature of the capsules should be above the production temperature of asphalt concrete. Figure 6a presented the mass loss curves of the raw materials of calcium alginate capsules. Figure 6b presented the thermal stability of calcium alginate capsules containing different healing agents. From 40 °C to 300 °C, the residual mass of calcium alginate capsules gradually reduced, which was attributed to the evaporation of free and bound moisture in the capsules and the breakdown of tiny amount of glycosidic bonds in the alginate chain. From 300 °C to 500 °C, the healing agents within the capsules gradually volatilized and masses of glycosidic bonds were decomposed and decarbonized, thus generating calcium carbonate and carbon dioxide. It can be seen that the three types of capsules containing different healing agents show similar thermal stability of before 180 °C (initial decomposition temperature of waste cooking oil).

As presented in Figure 7, the residual mass of vegetable oil capsule, waste cooking oil capsule and industrial rejuvenator capsule at 160 °C (production temperature of asphalt concrete) were 96.43%, 96.18% and 97.08%, respectively. The mass loss ratios of the three types of capsules at 160 °C were less than 5%, which implied that the capsules could resist the high temperature during the production process of asphalt concrete.

According to the mass loss ratios of the shell material calcium alginate, core materials (rapeseed oil, waste cooking oil and industrial rejuvenator) and different types of capsules in Figure 6, the healing agent contents within different capsules can be calculated according to Equations (2) and (3) [35].
(2)$$x+\left(y_{1}-y_{2}\right)\left(1-x\right)=Z_{1}-Z_{2}$$
(3)$$x=\frac{Z_{1}+y_{2}-y_{1}-Z_{2}}{1+y_{2}-y_{1}}$$
where *x_i_* is the rejuvenator content in capsules (%); *y*_1_ and *y*_2_ are the residual mass percentage of Ca-alginate at the initial decomposition temperature and complete decomposition temperature of the rejuvenator (%); *Z*_1_ and *Z*_2_ are the residual mass percentage of capsules at the initial decomposition temperature and complete decomposition temperature of the regulator (%).

The healing agent contents within different capsules containing vegetable oil, waste cooking oil, and industrial rejuvenator were 76.03%, 74.09% and 73.09%, respectively. These three types of capsules had higher healing agent content than the references [33,34,35,36], which was beneficial for self-healing of asphalt concrete containing the capsules.

### 3.4. Healing Properties of Differrnt Capsules under Different Cycles of Compression Loading

To investigate the effects of calcium alginate capsules containing different healing agents on the self-healing properties of asphalt concrete, the crack healing ratios of the specimens after different cycles’ compression loading of 1.4 MPa were tested and shown in Figure 8.

It can be seen in Figure 8 that, the strength recovery ratios of the asphalt concrete specimens without the capsules and with capsules containing vegetable oil, waste cooking oil and industrial rejuvenator, after 3 d healing at 20 °C (without compression loading) were 45%, 48.3%, 45.6% and 46.1%, respectively. There was not much difference between the strength recovery ratios of different specimens and the difference can be attributed to the slight healing agent released during the mixing and compaction of the mixture. The tiny difference in strength recovery ratio of test beams without and with capsules indicated that the capsules within asphalt concrete need to be activated by external stimulus to realize the self-healing function. The strength recovery ratios of the plain specimens after 5000, 10,000, 15,000 and 20,000 cycles of compression loading of 1.4 MPa were 52.5%, 53.6%, 55.2% and 56.1%, respectively, indicating that the compression loading cycles had small impact on the strength recovery ratios of the plain specimens. Increasing the number of loading cycles resulted in progressive compacting of the aggregates within the asphalt beams, as well as a modest reduction in the crack width between two cracked surfaces, both of which resulted in the slight uptrend of strength recovery ratios of test beams without capsules.

After different cycles of compression loading, all the fractured asphalt concrete specimens with the capsules containing different healing agents showed much higher strength recovery ratios than plain specimens. The strength recovery ratios of the specimens with the capsules containing different healing agents increased with the increase of the cycles of the compression loading. The reason was that more healing agent encapsulated in capsules would release and flowed into asphalt binder with the increase of loading cycles, thus improving the crack healing level [33,34]. After 5000, 10,000, 15,000 and 20,000 cycles’ compression loading, the strength recovery ratios of the specimens with the capsules containing vegetable oil were 59.0%, 68.0%, 76.2% and 82.8%, respectively, the strength recovery ratios of the specimens with the capsules containing waste cooking oil were 54.4%, 65.4%, 72.3% and 76.5% coma respectively, and the strength recovery ratios of the specimens with the capsules containing industrial rejuvenator were 56.0%, 66.3%, 75.6% and 80.6%, respectively. The results implied that the calcium alginate capsules containing different healing agents could enhance the self-healing property of asphalt concrete under cyclic compression loading. The capsules containing vegetable oil showed the best healing effect on asphalt concrete, while the capsules containing waste cooking oil showed the worst healing effect. The reason may be that fresh vegetable oil was composed of more light components while the light components within waste cooking oil was the least. More light components were conductive to the crack-healing and asphalt regeneration.

The fracture energy reflects the energy required to produce a new crack interface and can be used to evaluate the cracking resistance of asphalt concrete. The fracture energy recovery ratios of the fractured specimens after different cycles’ compression loading (1.4 MPa) and healing were tested and shown in Figure 9.

It can be seen in Figure 9 that with the increase of the compression loading cycles, the fracture energy recovery ratios of the fractured specimens containing different capsules increased significantly, while the increase of the fracture energy recovery ratios of the plain specimens without capsules was very limited. After 20,000 cycle’s compression loading, the fracture energy recovery ratios of the plain specimen increased from 38.5% to 47.6%. During the cyclic loading, the asphalt mixture beams were gradually compacted and the crack width between two cracked surfaces was decreased; thus, the fracture energy recovery ratios of beams without capsules increased slightly. The calcium alginate capsules can gradually release the rejuvenator inside under cyclic loading; therefore, the fracture energy recovery ratios of the fractured specimens containing different capsules were doubled after 20,000 cycle’s compression loading. Specifically, the specimens containing vegetable oil capsules showed the highest fracture energy recovery ratio of 96.6%. The fracture energy recovery ratios of the specimens containing waste cooking oil and industrial rejuvenator capsules were also as high as 90%. The fracture energy recovery ratios of the specimens with capsules were higher than that of specimen without capsules at the same conditions, which indicated that the rejuvenator released from the capsules during the cyclic loading rejuvenated the asphalt binder and thus increased the ductility and anti-cracking properties of the specimens.

### 3.5. Healing Properties of Different Capsules under Cyclic Compression Loading of Different Stress Levels

The tire pressure of the vehicles on the pavement can be different and will influence the rejuvenator release ratios and healing properties of different types of capsules within asphalt concrete. In this section, the healing properties of the fractured asphalt beams containing different capsules after 10,000 cycles of compression loading of different stress levels (0.6 MPa, 1.0 MPa and 1.4 MPa) were investigated.

The strength recovery ratios of the fractured asphalt beams after 10,000 cycles’ compression loading of different stress levels were shown in Figure 10. The strength recovery ratio of the plain asphalt beams increased slightly with the increase of the stress level. The increase of stress level would result in further compacting of the aggregates within the asphalt mixture beams, and cause a modest reduction in the crack width between two cracked surfaces, which jointly result in the slight uptrend of the fracture energy recovery ratios of beams without capsules. The strength recovery ratio of the plain asphalt beam without compression loading was 45.0%, which was attributed to the natural healing potential of asphalt. After 10,000 cycles’ compression loading of 0.6 MPa, 1.0 MPa and 1.4 MPa, the strength recovery ratios of the plain asphalt beam were 44.5%, 47.8% and 53.6%, respectively, and the increase of the strength recovery ratio was attributed to the reduction of the crack size caused by the compression loading. The strength recovery ratio of the asphalt beams containing the capsules increased more significantly with the increase of the stress level. After 10,000 cycles’ compression loading of 0.6 MPa, 1.0 MPa and 1.4 MPa, the strength recovery ratios of the samples containing vegetable oil capsules were 48.3%, 53.8%, 59.9% and 68.0%, respectively, the strength recovery ratios of the samples containing waste cooking oil capsules were 45.6%, 49.7%, 59.8%, 65.4%, respectively, and the strength recovery ratios of the samples containing industrial rejuvenator capsules were 46.1%, 50.1%, 60.5% and 66.3%, respectively. With the increase of the stress level of the cyclic compression loading, the elastic calcium alginate capsules within the asphalt concrete were suffered more deformation and, thus, released more rejuvenators, which improved the crack-healing efficiency and, thus, increased the strength recovery ratio of the specimens.

The fracture energy recovery ratios of the fractured asphalt beams containing different capsules after 10,000 cycles of compression loading at different stress levels were shown in Figure 11. It can be seen that the stress level showed limited effect on the fracture energy recovery ratios of the fractured asphalt beams without capsules. After 10,000 cycles’ compression loading of 0 MPa, 0.6 MPa, 1.0 MPa and 1.4 MPa, the fracture energy recovery ratios of the asphalt beams without capsules were 38.5%, 42.5%, 46.0% and 44.7%, respectively, while the fractured asphalt beams containing three types of capsules all showed much higher fracture energy recovery ratios after compression loading of 0.6 MPa, 1.0 MPa and1.4 MPa than without compression loading.

After 10,000 cycles’ compression loading of 0 MPa, 0.6 MPa, 1.0 MPa and 1.4 MPa, the fracture energy recovery ratios of the beams containing vegetable oil capsules were 42.3%, 58.3%, 70.4% and 76.0%, respectively, the fracture energy recovery ratios of the beams containing waste cooking oil capsules were 41.0%, 50.9%, 60.8% and 72.3% respectively, and the fracture energy recovery ratios of the beams containing industrial oil capsules were 40.1%, 55.2%, 68.6% and 74.8% respectively. The increase of the stress level of the compression loading improved the fracture energy recovery ratios of asphalt beams containing three types of capsules, which can be attributed to the fact that more rejuvenator was released at higher loading stress level.

In summary, in terms of strength recovery ratio and fracture energy recovery ratio, the vegetable oil capsules obtained high healing level, which further demonstrated the potential of vegetable oil self-healing capsules reported in the previous literatures [32,36,38].

### 3.6. Effect of Different Capsules on the SASA Fractions of Asphalt Binders Extracted from Asphalt Mixture Beams after Compressive Loading

Asphalt binders were extracted from the specimens after 10,000 cycles’ compression loading at 0.6 MPa, 1.0 MPa, 1.4 MPa respectively. The SRAR fractions of the extracted asphalt binders were shown in Figure 12. It can be seen that when no compressive loading was applied on test beams, the contents of light components (aromatics and saturates) of asphalt binder with the capsules were slightly higher than that of asphalt binder without capsules. The reason could be that the capsules released a small amount of rejuvenator during the mixing and compaction process of asphalt mixtures. The premature released rejuvenator would slightly improve the content of light components within asphalt binder. Moreover, without cyclic loading conducting on test beams, the contents of light components of asphalt binder with the three types of capsules showed little difference, which implied that the release of rejuvenator embedded in capsules was not significant after mixing and compaction of the asphalt mixtures. It also demonstrated that the obvious release of rejuvenator in capsules needed to be activated by the external compression loading.

Compared with plain asphalt within asphalt mixtures without capsules, the contents of light components within asphalt binder extracted from asphalt mixtures containing different capsules all increased with the loading stress level. The reason was that the higher loading pressure would induce the capsules release more healing agent and thus enhance the content of light components within asphalt binder. Moreover, when the loading stress level was constant, the content of light components within asphalt binder containing vegetable oil capsules was the highest, while the content of light component within asphalt binder containing waste cooking oil capsules was the lowest, which was consistent with the healing trend of asphalt concrete with the capsules. For instance, after 10,000 cycles of loading at 1.4 MPa, the light components content of asphalt binder within asphalt mixtures containing vegetable oil capsules, waste cooking oil capsules and industrial rejuvenator capsules were 71.06%, 63.37% and 68.49%, respectively. As shown in Table 1, the vegetable oil has the highest light components content while the waste cooking oil owned the lowest light components content. The difference of light components of three types of healing agent results in the content difference of light components within asphalt binders containing different capsules.

### 3.7. Effect of Different Capsules on the Rehological Property of Asphalt Binder Extracted from Asphalt Mixture Beams after Compressive Loading

Complex shear modulus (G*) characterizes the resistance to shear deformation of asphalt. Phase angle (δ) characterizes the elastic-viscous components ratio in the asphalt. These two parameters mutually describe the viscoelastic behavior of asphalt binder. The asphalt binder with lower G* and higher δ has a better flow ability, which is conductive to the healing of cracked asphalt.

The G* of the extracted asphalt binders from asphalt mixtures after loading of 0.6 MPa and 1.4 MPa were shown in Figure 13 and Figure 14, respectively. It can be seen that compared with virgin asphalt, the G* of all the extracted asphalt binders from the test beams with different capsules reduced obviously after different loading stress levels, the reason was that the healing agent released from the capsules could soften the asphalt and thus enhance the flow ability of asphalt binder.

For the specific capsules, the G* of the extract asphalt binder reduced with the increase of loading stress level. The reason was that more healing agent would be released due to higher loading stress applying on the asphalt beams containing the capsules, and more light components would be flowed into asphalt binder and, thus, reduce the viscosity of asphalt in crack zone, which was proven by the SARA results (Figure 12) in Section 3.6. Moreover, the asphalt binder extracted from asphalt mixtures with vegetable oil capsules has lowest G*, while the asphalt binder extracted from asphalt mixtures with waste cooking oil capsules has highest G*. For instance, after 10,000 cycles of loading at 1.4 MPa, the G*(46 °C) of asphalt binder within asphalt mixtures containing vegetable oil capsules, waste cooking oil capsules and industrial rejuvenator capsules were 17,808 Pa, 21,587 Pa and 19,356 Pa, respectively. The reason was that the content of light components in vegetable oil was higher than of waste cooking oil and industrial rejuvenator. More light components in asphalt binder would improve the flow and diffusion ability of asphalt, which would notably reduce the viscosity of asphalt.

In general, compared with virgin asphalt, the δ of all the extracted asphalt binder from the test beams with different capsules after different loading stress level increased, which was because the healing agent released from capsules could soften the asphalt and thus enhance the flow and self-healing ability of asphalt binder. Moreover, the asphalt binder extracted from asphalt mixtures with vegetable oil capsules has highest δ, while the asphalt binder extracted from asphalt mixtures with waste cooking oil capsules has lowest δ, which implied that the vegetable oil capsules has most significant enhancement on the flow ability of asphalt.

In summary, compared with asphalt mixtures without capsules, the incorporation of different capsules could soften the asphalt binder and improve its flow ability within asphalt mixtures after compressive loading.

## 4. Conclusions

In this study, calcium alginate capsules containing different healing agents were fabricated and their basic properties were characterized. The self-healing levels of asphalt mixtures with different capsules after various cycles of compressive loading and different loading stress levels were evaluated respectively. The SARA fractions and rheological properties of extracted asphalt binder from asphalt mixtures containing different capsules after different loading conditions were investigated. Based on the results, the following conclusion were drawn:The calcium alginate capsules containing vegetable oil, waste cooking oil and industrial rejuvenator presented obvious multi-cavity structure. The mechanical resistance and thermal stability of the three types of capsules all meet the requirement of mixing and compaction of asphalt mixtures in laboratory;The healing levels of asphalt mixtures with the three types of capsules all increased with the loading cycles and loading stress level. After 20,000 cycles of loading, the strength recovery ratios of test beams with vegetable oil capsules, waste cooking oil capsules and industrial rejuvenator capsules were 82.8%, 72.3% and 80.6%, respectively, which were notably higher than that of test beams without capsules (56.1%). Waste cooking oil can be used as the healing agent of calcium alginate capsules to improve the self-healing property of asphalt mixture;The contents of light components within asphalt binders extracted from asphalt mixtures containing different capsules all increased with the loading stress level. At the fixed loading pressure, the content of light components within asphalt binder containing vegetable oil capsules was higher than that of asphalt binder containing waste cooking oil and industrial rejuvenator capsules. After 10,000 cycles of loading at 1.4 MPa, the light components content of asphalt binder within asphalt mixtures containing vegetable oil capsules, waste cooking oil capsules and industrial rejuvenator capsules were 71.06%, 63.37% and 68.49%, respectively;The healing agents released from different capsules decreased the complex modulus and increase the phase angle of asphalt binder, thus improving the flow and self-healing ability of asphalt binder. After 10,000 cycles of loading at 1.4 MPa, the G*(46 °C) of asphalt binder within asphalt mixtures containing vegetable oil capsules, waste cooking oil capsules and industrial rejuvenator capsules were 17,808 Pa, 21,587 Pa and 19,356 Pa, respectively. The extracted asphalt binder within asphalt mixtures containing vegetable oil capsules had better flow ability than vegetable oil capsules and industrial rejuvenator.

This paper investigated the healing properties of asphalt concrete with capsules containing different healing agents after various cycles of loading and different loading pressures. The effect of loading pressure on the self-healing capacity of calcium alginate capsules were explored for the first time. The capsules with different healing agents all improve the healing level of asphalt concrete through releasing the healing agent, supplying the light components and softening the asphalt binder. This work provides a significant guide for the selection of healing agent for self-healing capsules in the future. The waste cooking oil can be a promising healing agent encapsulated in capsules. However, this work only explored the healing level of fresh asphalt concrete with different capsules. The effectiveness of these capsules on asphalt mixtures with various ageing levels will be explored in the future.

## Figures and Tables

**Figure 1 materials-15-05555-f001:**
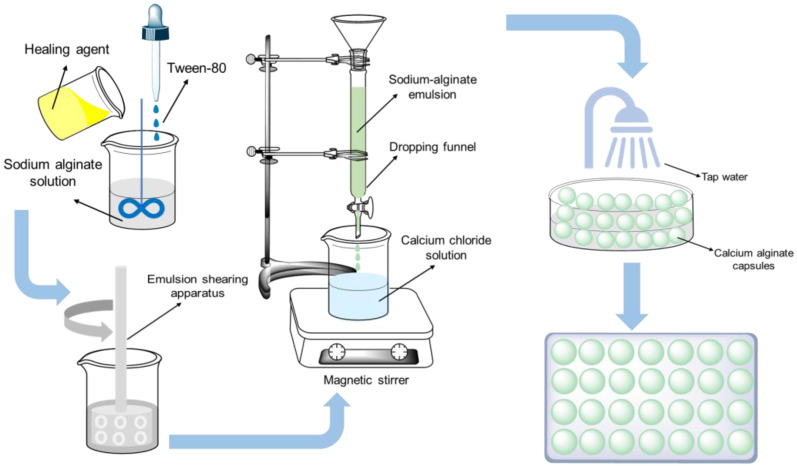
The fabrication procedure of calcium alginate capsules.

**Figure 2 materials-15-05555-f002:**
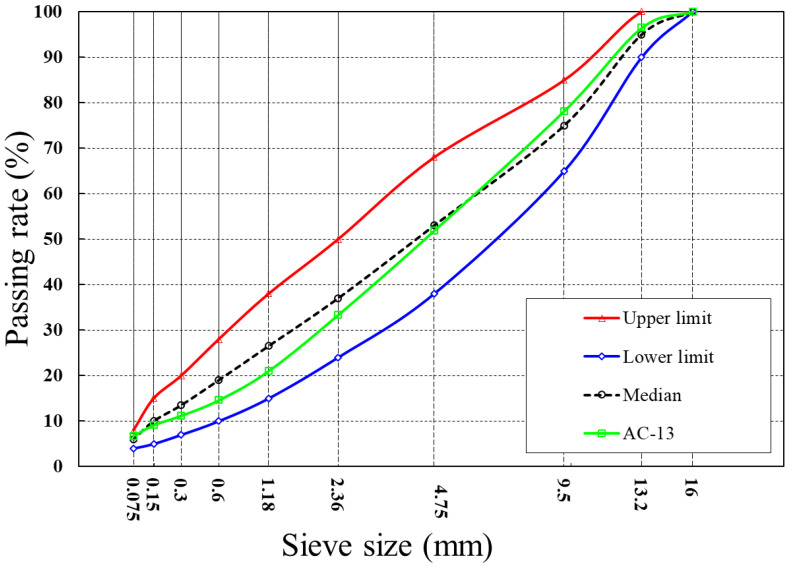
The aggregate gradation of AC-13 mixtures.

**Figure 3 materials-15-05555-f003:**
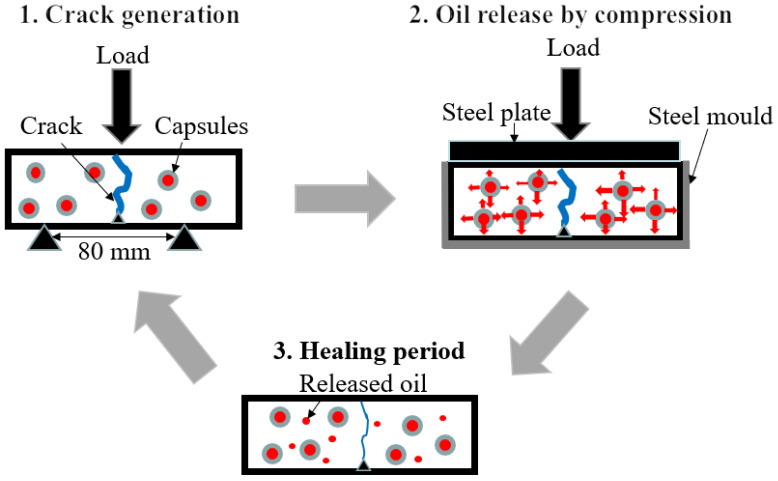
Fracture-cyclic compression loading–healing test procedure.

**Figure 4 materials-15-05555-f004:**
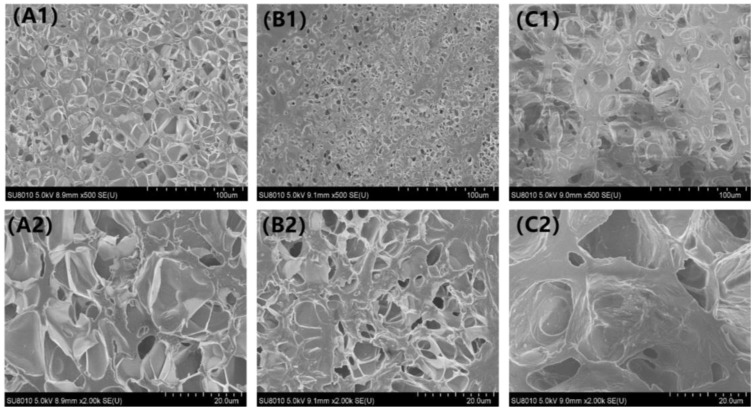
SEM images of capsules: (**A1**,**A2**) vegetable oil capsule, (**B1**,**B2**) waste cooking oil capsule and (**C1**,**C2**) industrial rejuvenator capsule.

**Figure 5 materials-15-05555-f005:**
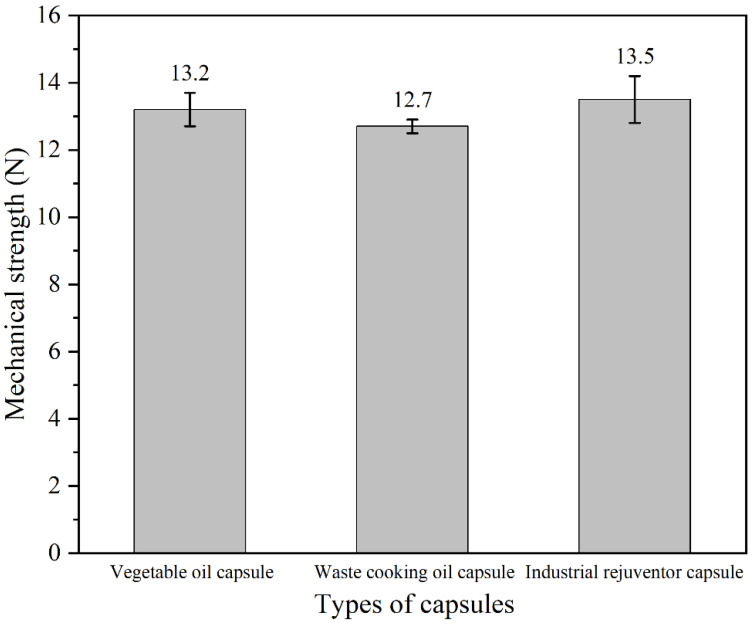
The mechanical strength of capsules containing different healing agent.

**Figure 6 materials-15-05555-f006:**
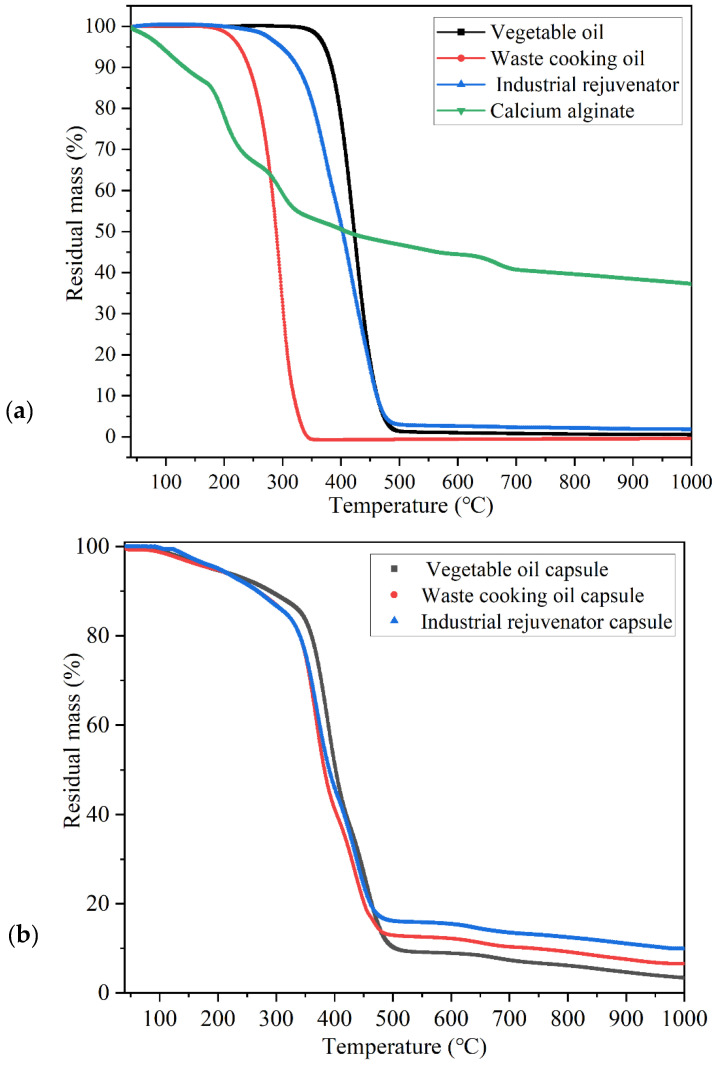
The mass loss curves of (**a**) raw materials and (**b**) capsules containing different healing agent.

**Figure 7 materials-15-05555-f007:**
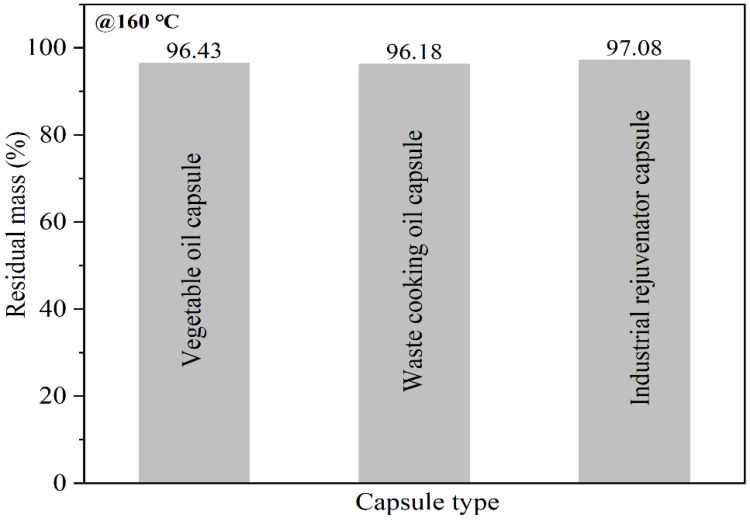
The residual mass of different capsules at 160 °C.

**Figure 8 materials-15-05555-f008:**
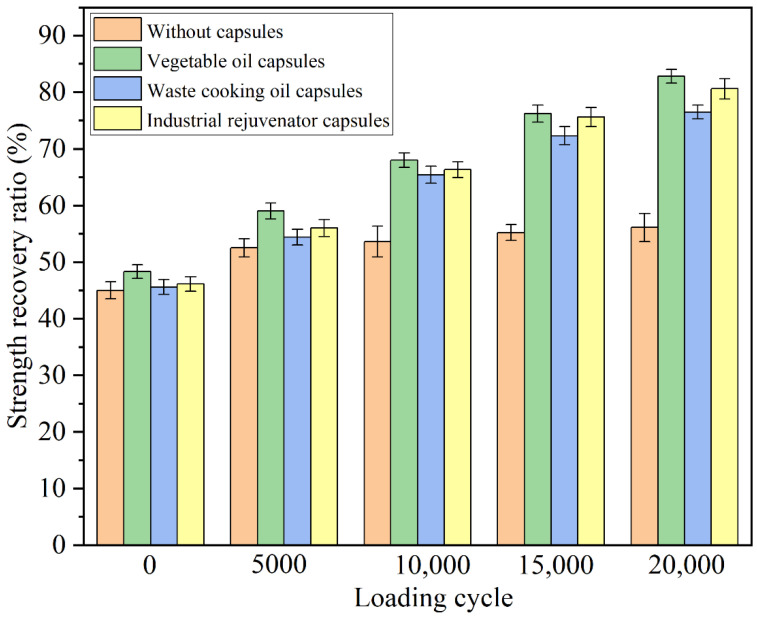
Strength recovery ratios of the fractured specimens containing different capsules after different cycles of compression loading at 1.4 MPa.

**Figure 9 materials-15-05555-f009:**
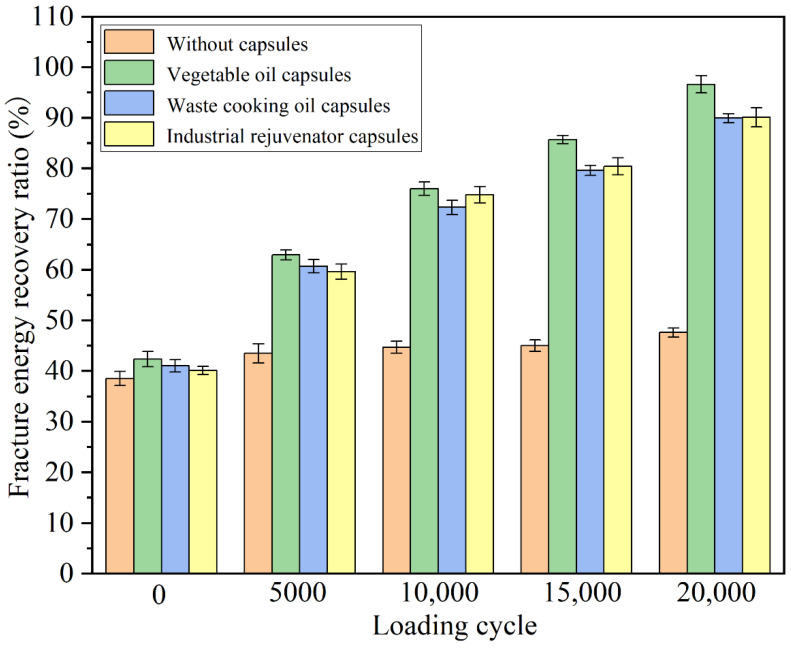
Fracture energy recovery ratios of the fractured specimens containing different capsules after different cycles of compression loading at 1.4 MPa.

**Figure 10 materials-15-05555-f010:**
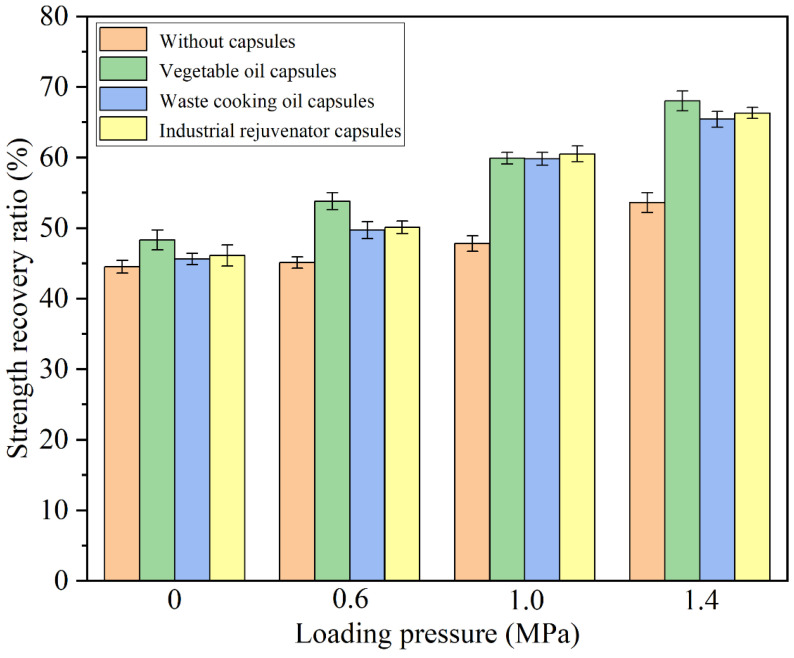
Strength recovery ratios of the specimens containing different capsules under 10,000 cycles’ compression loading of different stress levels.

**Figure 11 materials-15-05555-f011:**
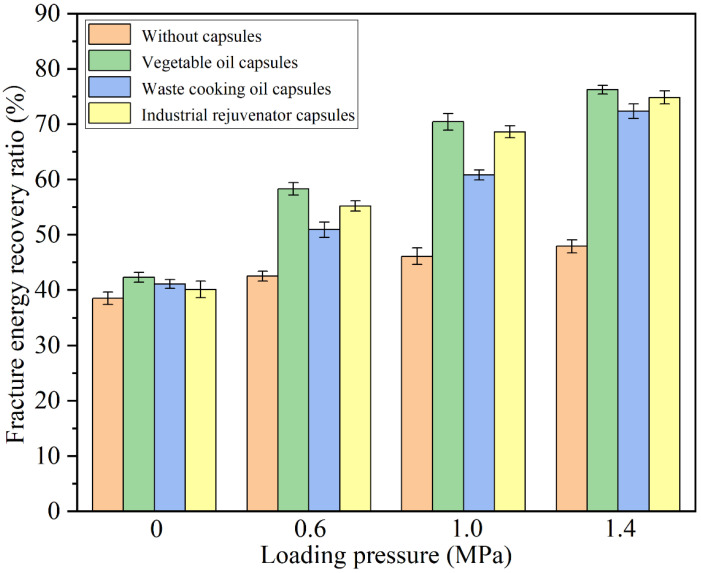
Fracture energy recovery ratios of the specimens containing different capsules under 10,000 cycles’ compression loading of different stress levels.

**Figure 12 materials-15-05555-f012:**
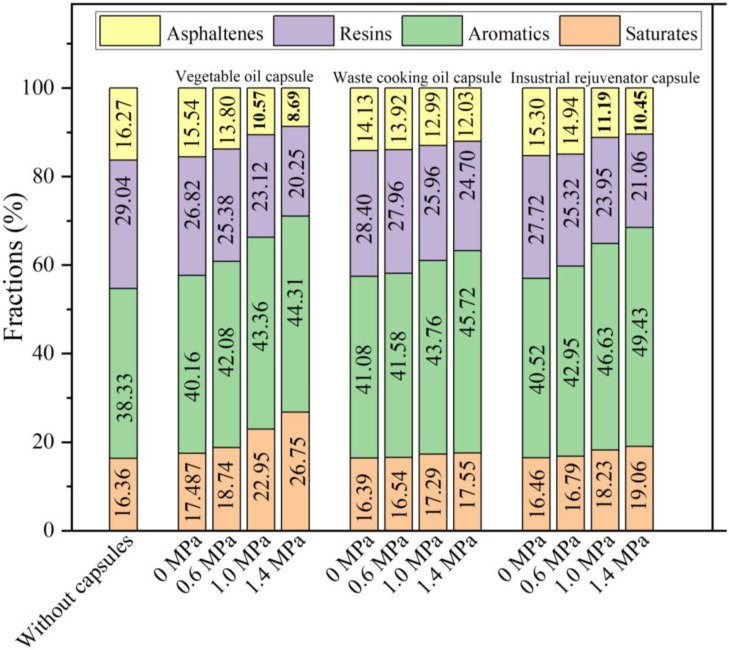
The SRAR fractions of the extracted asphalt binders.

**Figure 13 materials-15-05555-f013:**
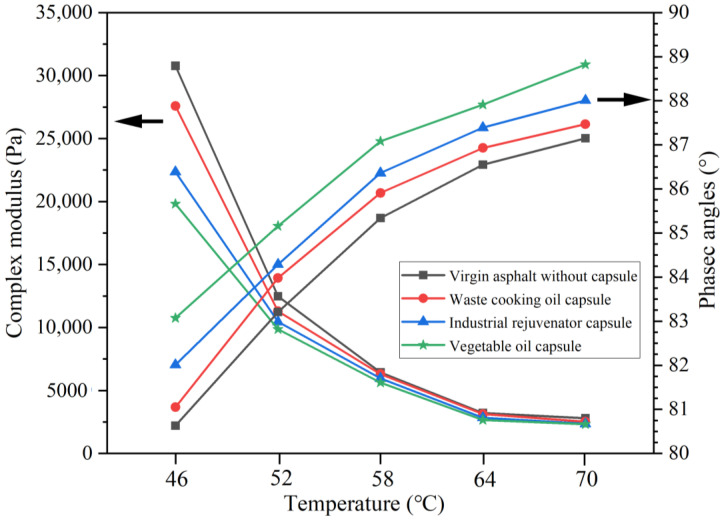
The G* and δ of the extracted asphalt binders within asphalt mixtures containing different capsules after loading at 0.6 MPa.

**Figure 14 materials-15-05555-f014:**
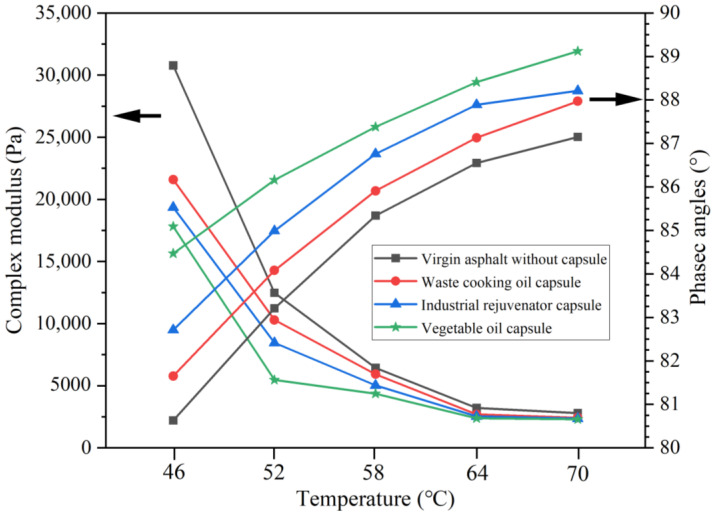
The G* and δ of the extracted asphalt binders within asphalt mixtures containing different capsules after loading at 1.4 MPa.

**Table 1 materials-15-05555-t001:** Properties of the three healing agents.

Healing Agents	Density (g·cm^−3^ (15 °C))	Viscosity (Pa·s (60 °C))	Flash Point (°C)	Saturates (%)	Aromatics (%)
Vegetable oil	0.935	0.28	320	61.9	38.1
Waste cooking oil	0.989	0.32	216	4.3	65.1
Industrial rejuvenator	0.980	0.43	220	6.0	75.0

## Data Availability

Data is contained within the article.

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
