# Peer review of "Self-Healing Properties of Asphalt Concrete with Calcium Alginate Capsules Containing Different Healing Agents"

_materials, 2022, doi:10.3390/ma15165555_

Round 1

Reviewer 1 Report

Dear Authors, thank you for you manuscript. It is quite interesting research. Few minor remarks: 1) intro and abstract have contain info on the novelty of your study, 2) couple of more references could be added in intro to make your intro more critical for comparison of what has been done before, 3) results discussion is quite general, 4) some main numerical results have to be reflected in the conclusions. 

Author Response

Dear Authors, thank you for your manuscript. It is quite interesting research. Few minor remarks:

Response: The authors would like to thank the reviewer for the valuable comments that helped to improve the quality of this manuscript. The manuscript has been revised according to the comments.

1) intro and abstract have contain info on the novelty of your study.

Response: Thank you for the valuable comment. We have highlighted the novelty of this paper in the abstract and introduction.

“The effects different healing agents on the self-healing performance of asphalt concrete has not been considered. In view of this, the paper aimed at exploring the self-healing properties of asphalt mixtures with calcium alginate capsules containing different healing agents.”

2) couple of more references could be added in intro to make your intro more critical for comparison of what has been done before.

Response: Thank you for the comment. We have added some references and revised the references carefully to void bulk citations.

3) results discussion is quite general.

Response: Thank you for the constructive comment. We have added explanations for the observed result changes and gave extended discussion on the obtained results.

Line 282-285: “Increasing the number of loading cycles resulted in progressive compacting of the aggregates within the asphalt beams, as well as a modest reduction in the crack width between two cracked surfaces, both of which resulted in the slight uptrend of strength recovery ratios of test beams without capsules.”

Line 290-292: “The reason was that more healing agent encapsulated in capsules would release and flowed into asphalt binder with the increase of loading cycles, thus improving the crack healing level [33, 34].”

Line 305-306: “More light components were conductive to the crack-repair and asphalt regeneration.”

Line 346-350: “The increase of stress level would result in further compacting of the aggregates within the asphalt mixture beams, and cause a modest reduction in the crack width between two cracked surfaces, which jointly result in the slight uptrend of the fracture energy recovery ratios of beams without capsules.”

Line 388-391: “In summary, in term of strength recovery ratio and fracture energy recovery ratio, the vegetable oil capsules obtained high healing level, which further demonstrated the potential of vegetable oil self-healing capsules reported in the previous literatures [36, 37].”

Line 420-427: “For instance, after 10000 cycles of loading at 1.4MPa, the light components content of asphalt binder within asphalt mixtures containing vegetable oil capsules, waste cooking oil capsules and industrial rejuvenator capsules were 71.06%, 63.37% and 68.49% respectively. As shown in Table 1, the vegetable oil has highest light components content while the waste cooking oil owned lowest light components content. The difference of light components of three types of healing agent results in the content difference of light components within asphalt binder containing different capsules.”

Line 451-454: “For instance, after 10,000 cycles of loading at 1.4MPa, the G*(46℃) of asphalt binder within asphalt mixtures containing vegetable oil capsules, waste cooking oil capsules and industrial rejuvenator capsules were 17,808 Pa, 21,587 Pa and 19,356 Pa respectively.”

4) some main numerical results have to be reflected in the conclusions. 

Response: Thank you for the comment. We have added numerical results in the conclusion part in the revised manuscript.

“After 20,000 cycles of loading, the strength recovery ratios of test beams with vegetable oil capsules, waste cooking oil capsules and industrial rejuvenator capsules were 82.8%, 72.3% and 80.6% respectively, which were notably higher than that of test beams without capsules (56.1%).”

“After 10000 cycles of loading at 1.4MPa, the light components content of asphalt binder within asphalt mixtures containing vegetable oil capsules, waste cooking oil capsules and industrial rejuvenator capsules were 71.06%, 63.37% and 68.49% respectively.”

“After 10,000 cycles of loading at 1.4MPa, the G*(46℃) of asphalt binder within as-phalt mixtures containing vegetable oil capsules, waste cooking oil capsules and industrial rejuvenator capsules were 17,808 Pa, 21,587 Pa and 19,356 Pa respectively”

Reviewer 2 Report

The article is about self-healing properties of asphalt concrete with calcium alginate capsules containing different healing agents. However, some issues must to be addressed:

  1. Abstract: Please start by expressing the aim of this paper, followed by the rest of the information. Also, please define or try to avoid using abbreviations in the abstract. Typically, the abstract should provide a broad overview of the entire project, summarize the results, and present the implications of the research or what it adds to its field.

2.    The bibliographic foundation is important and well executed, however some new discussions should be inserted, authors should consider some works in the literature, such as: DOI 10.1088/1757-899X/374/1/012019.

3.    Please update the introduction and reference sections with newest articles from 2022, not only self-citation …!

4.    Please avoid bulk citation like 1-4, 5-7, 8-12 …

5.    Line 137-138: please be more specific regarding “specification”.

6.    Figure 4 has no scientific soundness: please remove it!

7.    Please explain the novelty and the differences between this article and reference no. 27.

  1. The results are merely presented, not properly discussed. Please add explanations for the observed changes. Please give an extended discussion on the obtained results and correlate your findings with previous literature studies and prospective applications.
  2. More analysis and interpretation of the results should be added for a clearer understanding of observed experimental phenomena.
  3. The authors must to provide some details about importance of the research and their applicability.
  4. Please rewrite the conclusions in a more quantitative form and enhance the clarity of the conclusion section in order to highlight the results obtained.
  5. General check-up and correction of the English language is suggested. There are still some minor typos and grammatical errors.

The author needs to address the abovementioned points for the betterment of the manuscript.

Author Response

  1. Abstract: Please start by expressing the aim of this paper, followed by the rest of the information. Also, please define or try to avoid using abbreviations in the abstract. Typically, the abstract should provide a broad overview of the entire project, summarize the results, and present the implications of the research or what it adds to its field.

Response: Thank you for the constructive comment. In the revised manuscript, we have highlighted the aim of this work: “the paper aimed at…..”

Meanwhile, we have added the full name of abbreviation: saturates, aromatics, resins and asphaltenes (SARA).

Besides, we have presented the significance of the research in abstract: “This work provides a significant guide for the selection of healing agent for self-healing capsules in the future.”

  1. The bibliographic foundation is important and well executed, however some new discussions should be inserted, authors should consider some works in the literature, such as: DOI 10.1088/1757-899X/374/1/012019.

Response: Thank you for the kind suggestion. We have added the reference in the revised paper.

[10] D.D. Burduhos Nergis, M.M.A.B. Abdullah, P. Vizureanu, M.F.M. Tahir, Geopolymers and Their Uses: Review, IOP Conference Series: Materials Science and Engineering 374 (2018).

  1. Please update the introduction and reference sections with newest articles from 2022, not only self-citation …!

Response: Thank you for the kind reminder. Following your advice, we have reduced the self-citation and added some newest references (from 2022) into the revised manuscript.

[1] J. Tang, Y. Fu, T. Ma, B. Zheng, Y. Zhang, X. Huang, Investigation on low-temperature cracking characteristics of asphalt mixtures: A virtual thermal stress restrained specimen test approach, Construction and Building Materials 347 (2022).

[2] H. Liu, Z. Zhang, Z. Tian, C. Lu, Exploration for UV Aging Characteristics of Asphalt Binders based on Response Surface Methodology: Insights from the UV Aging Influencing Factors and Their Interactions, Construction and Building Materials 347 (2022).

  1. Please avoid like 1-4, 5-7, 8-12 …

Response: Thank you for the advice. Following your advice, we have rectified the references and citations to avoid citation pockets.

  1. Line 137-138: please be more specific regarding “specification”.

Response: Thank you for the kind reminder. We have added the code of the specification: JTG-F40-2004.

  1. Figure 4 has no scientific soundness: please remove it!

Response:Thanks for the comment. The SEM images in Figure 4 provided the interior structure of different capsules. The unique multi-cavity structure of the calcium alginate capsules with healing agent stored in different cavities was the structural cause of the gradual release of healing agent and long-term healing potential of the capsules. We think twice and decide to reserve this Figure.

  1. Please explain the novelty and the differences between this article and reference no. 27.

Response: Thank you for the comment. This article explored the healing properties of asphalt concrete with mm sized multi-cavity structure capsules containing different healing agents and explain the difference of healing level of three types of capsules in terms of rheological property and SARA fractions. This article provided strong evidence for vegetable oil as promising asphalt rejuvenator used in self-healing capsules. The reference no.27 investigated the effect of micrometer sized core-shell structure urea-formaldehyde microcapsules containing rejuvenator on the rheological property and healing ratio of asphalt binder under different stages. The structure and size of the capsules used in this research and the reference 27 are different.

  1. The results are merely presented, not properly discussed. Please add explanations for the observed changes. Please give an extended discussion on the obtained results and correlate your findings with previous literature studies and prospective applications.

Response: Thank you for the constructive comment. We have added explanations for the observed result changes and gave extended discussion on the obtained results. we also correlated our work with previous literature studies in the revised manuscript.

Line 282-285: “Increasing the number of loading cycles resulted in progressive compacting of the aggregates within the asphalt beams, as well as a modest reduction in the crack width between two cracked surfaces, both of which resulted in the slight uptrend of strength recovery ratios of test beams without capsules.”

Line 290-292: “The reason was that more healing agent encapsulated in capsules would release and flowed into asphalt binder with the increase of loading cycles, thus improving the crack healing level [33, 34].”

Line 305-306: “More light components were conductive to the crack-repair and asphalt regeneration.”

Line 346-350: “The increase of stress level would result in further compacting of the aggregates within the asphalt mixture beams, and cause a modest reduction in the crack width between two cracked surfaces, which jointly result in the slight uptrend of the fracture energy recovery ratios of beams without capsules.”

Line 388-391: “In summary, in term of strength recovery ratio and fracture energy recovery ratio, the vegetable oil capsules obtained high healing level, which further demonstrated the potential of vegetable oil self-healing capsules reported in the previous literatures [36, 37].”

Line 420-427: “For instance, after 10000 cycles of loading at 1.4MPa, the light components content of asphalt binder within asphalt mixtures containing vegetable oil capsules, waste cook-ing oil capsules and industrial rejuvenator capsules were 71.06%, 63.37% and 68.49% respectively. As shown in Table 1, the vegetable oil has highest light components content while the waste cooking oil owned lowest light components content. The difference of light components of three types of healing agent results in the content difference of light components within asphalt binder containing different capsules.”

Line 451-454: “For instance, after 10,000 cycles of loading at 1.4MPa, the G*(46℃) of asphalt binder within asphalt mixtures containing vegetable oil capsules, waste cooking oil capsules and industrial rejuvenator capsules were 17,808 Pa, 21,587 Pa and 19,356 Pa respectively.”

Line 517-519: “This work provides a significant guide for the selection of healing agent for self-healing capsules in the future. The waste cooking oil can be a promising healing agent encapsulated in capsules.”

  1. More analysis and interpretation of the results should be added for a clearer understanding of observed experimental phenomena.

Response: Thank you for the valuable comment. We have added more analysis and specific interpretation of the results in the revised manuscript. See the response to comment No. 8.

  1. The authors must to provide some details about importance of the research and their applicability.

Response: Thank you for the nice comment. We have highlighted the significance and applicability of this work.

In conclusion Line 515-519: “The effect of loading pressure on the self-healing capacity of calcium alginate capsules were explored for the first time.”  “This work provides a significant guide for the selection of healing agent for self-healing capsules in the future. The waste cooking oil can be a promising healing agent encapsulated in capsules.”

  1. Please rewrite the conclusions in a more quantitative form and enhance the clarity of the conclusion section in order to highlight the results obtained.

Response: Thank you for the insightful comment. We have added quantitative analysis of results in the conclusions

“After 20,000 cycles of loading, The the strength recovery ratios of test beams with vegetable oil capsules, waste cooking oil capsules and industrial rejuvenator capsules were 82.8%, 72.3% and 80.6% respectively, which were notably higher than that of test beams without capsules (56.1%).”

“After 10000 cycles of loading at 1.4MPa, the light components content of asphalt binder within asphalt mixtures containing vegetable oil capsules, waste cooking oil capsules and industrial rejuvenator capsules were 71.06%, 63.37% and 68.49% respectively.”

“After 10,000 cycles of loading at 1.4MPa, the G*(46℃) of asphalt binder within as-phalt mixtures containing vegetable oil capsules, waste cooking oil capsules and industrial rejuvenator capsules were 17,808 Pa, 21,587 Pa and 19,356 Pa respectively”

  1. General check-up and correction of the English language is suggested. There are still some minor typos and grammatical errors.

Response: Thank you for the kind reminder. We have checked the typo and grammar of language carefully and made corresponding corrections.

Round 2

Reviewer 2 Report

The article is suitable for publication.